# STL1, a New AKT Inhibitor, Synergizes with Flavonoid Quercetin in Enhancing Cell Death in A Chronic Lymphocytic Leukemia Cell Line

**DOI:** 10.3390/molecules26195810

**Published:** 2021-09-25

**Authors:** Carmen Cervellera, Maria Russo, Serena Dotolo, Angelo Facchiano, Gian Luigi Russo

**Affiliations:** National Research Council, Institute of Food Sciences, 83100 Avellino, Italy; carmen.cervellera@isa.cnr.it (C.C.); maria.russo@isa.cnr.it (M.R.); sdotolo@unisa.it (S.D.); angelo.facchiano@isa.cnr.it (A.F.)

**Keywords:** autophagy, apoptosis, quercetin, AKT inhibitor

## Abstract

Using a pharmacophore model based on the experimental structure of AKT-1, we recently identified the compound STL1 (ZINC2429155) as an allosteric inhibitor of AKT-1. STL1, was able to reduce Ser473 phosphorylation, thus inhibiting the PI_3_K/AKT pathway. Moreover, we demonstrated that the flavonoid quercetin downregulated the phosphorylated and active form of AKT. However, in this case, quercetin inhibited the PI_3_K/AKT pathway by directly binding the kinases CK2 and PI_3_K. In the present work, we investigated the antiproliferative effects of the co-treatment quercetin plus STL1 in HG-3 cells, derived from a patient affected by chronic lymphocytic leukemia. Quercetin and STL1 in the mono-treatment maintained the capacity to inhibit AKT phosphorylation on Ser473, but did not significantly reduce cell viability. On the contrary, they activated a protective form of autophagy. When the HG-3 cells were co-treated with quercetin and STL1, their association synergistically (combination index < 1) inhibited cell growth and induced apoptosis. The combined treatment caused the switch from protective to non-protective autophagy. This work demonstrated that cytotoxicity could be enhanced in a drug-resistant cell line by combining the effects of different inhibitors acting in concert on PI_3_K and AKT kinases.

## 1. Introduction

Quercetin (3,3′,4′,5,7-pentahydroxyflavone) is a secondary metabolite of plants belonging to the class of flavonoids, the largest group among polyphenols. It is present in a variety of fruits (apples, grapes, olives, citrus fruits, berries), vegetables (tomatoes, onions, broccoli, capers), beverages (tea and red wine), and herbal extracts. Flavonoids count more than 4000 compounds and are characterized by a low molecular weight and a very wide distribution in nature [1]. In plants, flavonoids are synthesized from the aromatic amino acids phenylalanine and tyrosine. The biochemical modifications that depend on light and temperature lead to final compounds characterized by a typical structure with 15 carbon atoms, consisting of two benzene rings connected through a heterocyclic intermediate ring [2]. Quercetin is a flavonol, more precisely, a tetra-oxy-flavonol, also known as antoxanthins. The molecule (Figure 1a) is characterized by the presence of three rings, two of which are aromatic (A–B) and one heterocyclic with an oxygen atom (C). Quercetin, from a strictly chemical point of view, is defined as a reducing agent. In fact, the first works published in the early 1990s focusing on quercetin showed an effective in vitro antioxidant activity of the molecule [3]. These early studies led researchers to explain the effects of quercetin (and other polyphenols) on animal cells simply due to its antioxidant activity. In fact, in nature, flavonoids act as scavengers of reactive oxygen species (ROS) generated by the exposure to ultraviolet rays in plant tissues or are synthesized by plants in response to environmental stress factors, such as temperature variations or following of fungal and microbial attacks. Considering the pleiotropic functions in plant tissues, the different polyphenolic molecules, such as quercetin, can interact with multiple enzymes even in animal cells. Additionally, the research in recent years has been focusing above all on identifying the molecular targets of these compounds, in order to exploit their pharmacological potential in treating various human pathologies.

Analyzing the biological activities of quercetin in the context of all stages of carcinogenesis, from initiation to invasion and metastasis [8], emerged the ability of the molecule to interfere with different molecular targets identified as “hallmarks of cancer” [9,10], making it as a potential chemopreventive agent with pleiotropic effects against cancer. Our previous studies mostly focused on the capacity of quercetin to enhance cell death in apoptosis-resistant leukemic cells when the molecule was associated with different apoptotic inducers (e.g., the anti-CD95 agonist antibody, recombinant TRAIL, chemotherapeutic drugs, BH3-mimetics) [5,11]. In particular, we demonstrated that quercetin, at relatively low doses (10–20 µM) and in mono-treatment, was not cytotoxic. Additionally, it did not show any significant apoptotic effect on acute lymphoblastic leukemia cell lines (HPB-ALL and Jurkat) and in B cells isolated from chronic lymphocytic leukemia (CLL) patients, both cellular models being highly resistant to ABT-737, a BH3 mimetic. However, the co-treatment with quercetin and ABT-737 synergistically decreased resistance to apoptosis and induced massive cell death [6]. Subsequently, we proved that in HG-3 cells, also derived from a patient affected by CLL, quercetin enhanced apoptosis in the presence of ABT-737 by directly inhibiting the CK2 and PI_3_K protein kinases and, as a consequence, inactivating the PI_3_K/AKT/Mcl-1 signaling pathway that in CLL is often associated with drug resistance [4]. These findings may be relevant in terms of the potential clinical application considering that CLL remains the most common form of leukemia in Western countries. The overall survival (OS) at 5 years of high-risk patients, those resistant to therapy, is of about 20% [12]. Moreover, considering that, a significant group of patients with CLL survive for many years or decades in the absence of treatments (Rai stage 0 or Binet classification A) due to the relatively slow progression rate of the disease, a chemopreventive intervention with low doses of quercetin may result as beneficial [13].

Furthermore, we demonstrated that the inhibitory effect of quercetin on the PI_3_K/AKT pathway was associated with the downregulation of the phosphorylated and active form of AKT [4,6]. Protein kinase B/(PKB)/AKT was identified about 30 years ago by cloning the v-AKT oncogene. The over-activation of AKT, caused by the increased synthesis of the membrane lipid PIP_3_ (phosphatidyl-inositol-3,4,5-tris-phosphate), produced by PI_3_K, has been described in 50% of human cancers [14,15,16,17]. AKT exists in three isoforms, AKT-1 (PKBα), AKT-2 (PKBβ), and AKT-3 (PKBγ), encoded by three distinct genes. Each isoform is characterized by a conserved catalytic domain flanked by an N-terminal domain, the PH (Pleckstrin Homology), and a C-terminal regulatory domain (EXT) that includes the hydrophobic motif (HM), typical of the ACG kinases [18]. When the PH domain binds to PIP_3_, it undergoes conformational changes that favor the phosphorylation of residues Thr308 and Ser473 by PDK1 (3-phosphoinositide dependent protein kinase 1) and mTORC2 (mechanistic target of rapamycin (mTOR) complex 2) kinases, respectively, leading to the full activation of AKT. The signal is stopped by the lipid phosphatase PTEN that dephosphorylates PIP_3_ and by phosphatases PP2A, which dephosphorylates Thr308 and PHLPPs (PHLPP1 and PHLPP2), that remove the phosphate group from Ser473 [19].

Due to the fact that the PI_3_K/AKT pathway is frequently modified in human cancers, it became one of the favored targets for new and highly specific anticancer drugs designed to minimize cytotoxic side effects on normal cells [20]. As an example, idelalisib (CAL-101) targets the isoform δ of the catalytic subunit of PI_3_K (p110 δ) and it is effective against CLL, follicular lymphoma (FL), and small lymphocytic lymphoma (SLL) [21,22]. Duvelisib, approved in 2018 for the treatment of patients with relapsed or refractory CLL, is a dual inhibitor of PI3Kδ and PI3Kγ [23]. Other PI3Kδ are currently under development [12]. More intriguing is the synthesis and clinical efficacy of inhibitors that directly target AKT. In fact, the complexity and the central role of this kinase generates compensatory mechanisms, such as the hyper-activation of PI_3_K or mTOR in the PI_3_K/AKT/mTOR pathway. Therefore, to avoid the phenomena of compensatory resistance, a possible therapeutic strategy resides in the treatments with combinations of inhibitors targeting both AKT and/or either one between PI_3_K and mTOR [15].

In this scenario, we recently completed a computational screening to search for compounds that are able to bind the AKT protein by generating a pharmacophore model based on the experimental structure of AKT-1 in complex with IQO, a well-known AKT-1 inhibitor. We selected two compounds, ZINC2429155 (STL1) (Figure 1b) and ZINC1447881 (AC1) both suitable in silico to bind AKT-1 in an allosteric site, but only one, STL1, was able in vitro to reduce Ser473 phosphorylation, thus inhibiting the PI_3_K/AKT pathway [7]. However, the inactivation of AKT phosphorylation by STL1 was not sufficient to induce cell death in the malignant cell line investigated as represented by the HG-3 cells, a lymphoblastoid cell line with B1 cell characteristics established from a CLL clone by the in vitro EBV infection [24].

Based on these data, the present work aims to investigate the antiproliferative effects of the co-treatment quercetin plus STL1 in HG-3 cells. The rationale resides in the attempt to demonstrate that cytotoxicity can be enhanced in a drug-resistant cell line, such as HG-3, by combining the effects of inhibitors acting in concert on PI_3_K and AKT, such as quercetin and STL1, respectively, with the net result to block or at least weaken the pro-survival effect of PI_3_K/AKT pathway.

## 2. Results

### 2.1. Quercetin Enhances Cell Death Induced by STL1 in HG-3 Cells

Recently, we demonstrated that ZINC2429155 (also named STL1), a compound selected after the computational analysis, inhibited pAKT-Ser473 expression at the highest concentration applied (40 μM), without significantly affecting the HG-3 cell viability [7]. In parallel, previous works from our laboratory demonstrated that quercetin did not induce apoptosis per se. However, when associated with other pro-apoptotic drugs, quercetin increased cell death in apoptosis-resistant cell lines, including HG-3 [7,9,10]. Based on this rationale, we explored the possibility that the combined treatment between quercetin and STL1 could bypass resistance in HG-3 cells. As reported in Figure 2, cells treated with 5–20 μM STL1 or 2.5–10 μM quercetin, showed only a minimal, although statistically significant, reduction in cell viability, not higher than 10–15% compared to the control (DMSO-treated cells). However, when the two compounds were added in combination, cell viability was reduced to 40–50% (Figure 2a,b). The combination between quercetin and STL1 was synergic as demonstrated by the calculation of the combination index (CI) that resulted < 1 (Figure 2c).

In the following experiments, we applied the concentrations of 10 and 20 μM for quercetin and STL1, respectively, which provided more reproducible and coherent data.

### 2.2. Quercetin Enhances STL1 Sensitivity in HG-3 through AKT Inhibition

To explore the central role of PI_3_K/AKT in generating the synergistic and cytotoxic effect of quercetin plus STL1 in HG-3 cells (Figure 2), we measured the expression of the phosphorylated and active form of AKT (p-AKT^Ser473^) shortly after the combined treatment (1 h). As expected (Figure 3), both STL1 and quercetin in the mono-treatments were able to almost abolish the phosphorylation of AKT, the former acting directly on AKT [7], the latter inhibiting the upstream kinases CK2 and PI_3_K [4]. However, in both cases, the decreased expression of p-AKT^Sr473^ did not coincide with a parallel decrease in cell viability (Figure 2). Differently, in the co-treatment, the p-AKT^Ser473^ expression was also almost totally abolished, but in this case (Figure 3), extensive cell death was observed (Figure 2). The demonstration that the PI_3_K/AKT pathway was involved in this process was confirmed by the use of CAL-101, a canonical PI_3_K inhibitor used as a positive control (Figure 3).

### 2.3. Quercetin and STL1 Co-Treatment Induce Apoptosis in HG-3 Cells

To investigate the type of cell death induced by quercetin plus the STL1 co-treatment on HG-3 cells, we measured the presence of apoptotic nuclei. As reported in Figure 4a,b, the number of apoptotic nuclei significantly increased compared to the control (DMSO treated cells) only in the presence of the combined treatment, to indicate an evident activation of apoptosis that was also confirmed by changes in the caspase-3 activity. In fact, the early measurement of this enzymatic activity (6 h after mono- and combined treatments) indicated that only when both quercetin and STL1 were present, the level of caspase-3 significantly increased (Figure 4c).

### 2.4. Quercetin and STL1 Co-Treatment Induces Autophagy in HG-3 Cells

Based on the observation that quercetin induces protective autophagy in different cancer cell lines [25,26], we verified if the very limited apoptotic effect of quercetin in the mono-treatment (Figure 4) could be due to the activation of this specific form of autophagy. As reported in Figure 5a,b, the treatment with quercetin was slightly associated with the presence of autophagosomes, whose number significantly increased in the combined treatment. Two independent biochemical markers were measured to characterize the activation of autophagy. For LC3-II, the lipidated form of LC3, we observed an increased expression that paralleled with the appearance of autophagosomes (Figure 5c), as expected. More intriguing was the result of the expression of ATG5, a key protein responsible for the extension of the autophagic membrane in autophagic vesicles [27,28]. In this case, unexpectedly, we detected a decrease of about 50% after the combined treatment compared to the DMSO treated cells (Figure 5d). The reduced expression of ATG5 was in agreement with its pro-apoptotic role following the cleavage by caspases or calpaine proteases [29]. It is interesting to note that for both STL1 and quercetin mono-treatments, the expressions of LC3-II and ATG5 fluctuated in the direction of an increased autophagy and apoptosis, respectively (Figure 5c,d).

The observation that the autophagy activation in the presence of quercetin and STL1 mono-treatments was not associated with cell death, while the two compounds together induced apoptosis, suggested that their combination might promote the switch from the protective autophagy to the non-protective form. To verify this hypothesis, we measured the changes in cell viability in the presence of Bafilomycin A1 (BAF) and 3-Methyladenine (3-MA), two of the most adopted autophagy inhibitors [30]. For both quercetin and STL1, the addition of BAF or 3-MA significantly decreased cell viability, as expected, considering the autophagy-protective effect of the mono-treatments. More in detail, for quercetin, BAF and 3-MA reduced cell viability by 15.5 and 11.5%, respectively (Figure 6a,b; green bars). For STL1, BAF and 3-MA diminished cell viability by 16.4 and 25.2%, respectively (Figure 6a,b; grey bars). However, the presence of BAF or 3-MA in the combined treatments did not significantly change the percentage of cell death (Figure 6a,b; brown bars), suggesting that the co-treatment was associated with a non-protective form of autophagy.

## 3. Discussion

The present work describes the dual behavior of two inhibitors of the PI_3_K/AKT pathway, namely quercetin and STL1, in modulating autophagy and apoptosis in a CLL-derived cell line. According to the seminal review of Guido Kroemer’s group [31], autophagy (i.e., macroautophagy) and apoptosis can sometimes generate mixed phenotypes with autophagy that can favor cellular adaptation to external stressors, thus allowing the cells to recover. In this sense, autophagy can inhibit apoptosis. Alternatively, autophagy can temporarily block apoptosis and later activate a form of autophagic cell death also called type II cell death to distinguish it from the type I cell death, indicating the “canonical” apoptotic process with cellular shrinkage, pyknosis, and karyorrhexis. This crosstalk between autophagy and apoptosis is facilitated by intermediates common to pathways leading to either one of the two processes and by the existence of variable cellular thresholds that can influence the cellular decision to tend for autophagy or apoptosis [31]. Another important piece of information comes from Gewirtz’s works [32,33], who defined four different forms of autophagy: Protective (or cytoprotective), cytotoxic, cytostatic, non-protective. Limiting the discussion to cancer cells, extensive data indicate that protective autophagy represents the cells’ “defensive” response to anticancer drugs. When protective autophagy is blocked by pharmacological inhibitors (chloroquine, bafilomycin A1 or 3-methyladenine) or genetic inhibition, sensitivity to therapy and/or apoptosis are increased. The non-protective autophagy does not differ in intensity from other forms and its inhibition (pharmacological or genetic) does not influence sensitivity to therapy [32]. Gewirtz also introduced the concept of “autophagic switch” between the protective and non-protective forms of autophagy and its clinical significance [33,34]. Based on these key literature data, in the next paragraphs, we will try to interpret the different effects of quercetin and STL1 described in the present work on the HG-3 cell death and cell survival, when applied in mono-treatments or in combination.

The relatively low doses of quercetin and STL1 (10 and 20 μM, respectively) applied in our assays indicate that when added alone to HG-3 cells, they did not significantly influence cell proliferation (Figure 2) and, consequently, they do not activate apoptosis (Figure 4). The limited cytotoxicity observed in Figure 2 for both compounds (<10%) can be considered physiological and probably due to different and independent factors including the presence of an asynchronous population of HG-3 cells, with the possibility that cells in a specific phase of the cell division cycle could be more sensitive to the treatments.

Since the molecular targets of quercetin and STL1 in HG-3 cells are known, with quercetin being a direct inhibitor of CK2 and PI_3_K [4] and STL1 being an allosteric inhibitor of AKT [7], we hypothesized that the lack of effects in terms of reduced cell growth, expected for inhibitors of the PI_3_K/AKT pathway, could be due to multiple factors including: (i) Limited uptake and, consequently, low intracellular concentrations of quercetin and/or STL1 at the dosages applied; (ii) activation of “compensatory” and functionally “complementary” pathways that reduce and/or abrogate their inhibitory effects on the target kinases; (iii) activation of protective autophagy. The first two possibilities are not easy to demonstrate, although we cannot exclude them. However, certainly, the third hypothesis is mostly supported by data presented in Figure 6, where clearly the presence of two different autophagic inhibitors, BAF and 3-MA, induced a significant decrease in cell viability, as expected in the presence of cytoprotective autophagy [32].

Even more difficult is to explain the molecular mechanism(s) responsible for the dual effects, apoptosis (Figure 4) and autophagy (Figure 5), observed when quercetin and STL1 were combined. Firstly, it is important to highlight the pharmacological importance of this result, since the CI < 1 indicates a clear synergistic effect that needs to be taken into account for future studies, more oriented to investigate the potential therapeutic efficacy of this co-treatment. The results in Figure 6 (brown bars) suggest the presence of a non-protective form of autophagy, since its inhibition did not influence the sensitivity to the combined treatment, as expected according to [32]. This observation is even more remarkable considering that it was obtained using two different inhibitors acting on the early and late phase of the autophagic flux. In fact, 3-MA blocks the vesicle nucleation, when mammalian Vps34, a class III PI_3_K, assembles in a multiprotein complex, which includes Beclin-1, UVRAG (UV irradiation resistance-associated tumor suppressor gene), and Vps15, a myristylated kinase [35]. On the contrary, BAF, which is similar to chloroquine, inhibits the late phase of the autophagic process, when autophagosomes undergo maturation and fuse with lysosomes to create autolysosomes [36].

The key question remains: Who comes first, autophagy or apoptosis? Are they correlated or independent events? It is possible that the combination of quercetin plus STL1 activates both processes in parallel and independently from each other. There is also the possibility that apoptosis follows autophagy. Circumstantial evidence suggest that the latter possibility can occur in our system. In fact, the appearance of apoptotic nuclei (Figure 4) was observed after 48 h of the treatment, while the early autophagic marker, LC3-II, shown in Figure 5c was already detectable after 24 h from the co-treatment. In addition, the strong reduction in the expression of the full-length ATG5 band observed in Figure 5d suggests the calpain-dependent degradation of the protein with the generation of the truncated fragment of 24 kDa that, freed from autophagic functions, translocates to the mitochondria where it binds and inhibits Bcl-X_L_, contributing to the activation of apoptosis [37,38]. A corollary consequence of this interpretation is that the autophagic process here described is ATG5-independent or at least dispensable from the presence of ATG5 in assembling the autophagosome, a condition well characterized in the recent literature [39,40].

Alternatively, we might hypothesize a situation for the interaction between apoptosis and autophagy similar to the one well described by the already mentioned review from Maiuri et al. [31]. In this case, cellular stressors can trigger “an autophagic response that, independently of the autophagic response, is followed by apoptosis. In this scenario, the order of events (autophagy, then apoptosis) is chronological, not hierarchical, meaning that inhibition of autophagy does not prevent apoptosis”. If this is the case, future experiments will be designed in HG-3 cells co-treated with quercetin and STL1 to determine if the consequences of apoptosis inhibition will result in cell survival or necrosis.

A further issue to be debated is why the apoptotic effect linked or not to autophagy, appeared and was synergic only in the combined treatment, quercetin plus STL1, with respect to the mono-treatments. From what we discussed in the previous paragraphs and sections, both compounds insist on the PI_3_K/AKT pathway: Quercetin directly on PI_3_K and on one of its positive modulators, the kinase CK2 [4], while STL1 directly inhibits AKT [7]. The supposed existence in drug-resistant cells of different sensitivity thresholds, of still undefined nature, which govern the decision towards autophagy or apoptosis, suggests that when the effects of stressors remain below the threshold, mixed phenotypes can appear due to the possible reciprocal inhibition of the two processes with autophagy tending to adapt single cells to stressors [31]. Probably, when quercetin and STL1 are combined, conditions exist for a substantial inhibition of the PI_3_K/AKT pathway that exceeds the apoptosis threshold and/or induces non-protective autophagy by the inactivation of mTOR, that frees the ULK1 complex to initiate the process of autophagosome formation [41]. However, the observation that the two compounds exert a synergistic effect, leaves open the hypothesis that the amplification of the inhibitory signals may involve substrates different than the key kinases regulating the PI_3_K/AKT pathway. As an example, we discussed elsewhere [4] the possibility that quercetin can directly bind the BH3 domain of Bcl-2 and Bcl-X_L_, inhibiting their activity and promoting apoptosis [42]. Concerning STL1, we demonstrated that this inhibitor binds the AKT-1 isoform [7], but we do not know if and which other AKT isoforms are expressed in HG-3 cells and whether STL1 binds and inhibits them with an efficacy similar to AKT-1. The observation that other leukemia-derived cell lines, such as K562 and Meg-01, express all three AKT isoforms [43] stimulates the hypothesis that the STL1 inhibitory effect could be amplified if future works will demonstrate that other AKT isoforms are present in HG-3 and are triggered by STL1.

A final comment is devoted to the possible role of BH3 mimetics in our experimental model. As reported in the Introduction, we demonstrated that quercetin, at doses comparable to those used in the present work, synergistically potentiated the therapeutic response to the BH3 mimetic, ABT-737, in cell lines, including HG-3, and in B cells isolated from CLL patients [4,6]. ABT-737 is a strong inhibitor of several Bcl-2 family members, including Bcl-2, Bcl-X_L_, and Bcl-w [44], and some of them, e.g., Bcl-2, Bcl-X_L_, play a key role in blocking autophagy (type II cell death) by binding and inactivating Beclin-1 (the mammalian orthologue of yeast ATG6), whose interaction with Vps34, Vps15, UVRAG, Ambra1, and Bif-1 drives the initial steps of phagophore formation [45]. Therefore, in the next future, it will be of great interest to verify if the association between STL1 and ABT-737 (or other BH3 mimetics), in the presence or absence of quercetin, can result in a more potent therapeutic effect, inducing an autophagic switch from the cytoprotective to the cytotoxic form.

## 4. Materials and Methods

### 4.1. Reagents

The Roswell Park Medium Institute (RPMI) medium, L-glutamine 200 mM, penicillin 5000 IU/mL/streptomycin 5000 μg/mL were purchased from ThermoFisher Scientific/Life Technologies (Milan, Italy), as well as the fetal bovine serum from ThermoFisher Scientific/Life Technologies. Trypan blue solution (0.4% *v*/*v*), propidium iodide, quercetin, 3-methiladenine (3-MA), and dimetylsulfoxide (DMSO) were from Merck Life Science (Milano, Italy). Hoechst stain (33258) was from Enzo Life Science (distributed by Euroclone, Pero, Milan, Italy). ZINC2429155 (STL1) was purchased from Vitas-M Limited, Hong Kong (https://vitasmlab.biz accessed on 27 September 2021), dissolved in DMSO, aliquoted, and stored at −20 °C.

### 4.2. Cell Culture and Treatment

All of the experiments were performed employing the HG-3 cell line, one of the clones obtained after incubating with the B95–8 EBV virus, the mononuclear cell population derived from the blood of a IGHV1–2 unmutated CLL patient, as previously reported [24]. HG-3 cells possess B1 cell characteristics and, similar to other CLL cells, expressed CD5/CD20/CD27/CD43. They were cultured in the RPMI medium supplemented with 10% fetal bovine serum, 1% L-glutamine, and 1% penicillin/streptomycin at 37 °C in a humidified atmosphere containing 5% CO_2_. Cells were counted with the Trypan blue system performed using the EVE Automatic cell counter (NanoEntek distributed by VWR, Milan, Italy) to verify their viability before starting each experiment, usually >90%.

### 4.3. Cell Viability Assay and Combination Index Determination

Cell viability was assayed by the CyQuant reagent (ThermoFisher Scientific/Life Technologies) to quantify the number of living cells, using a nuclear dye that selectively binds to nucleic acids, emitting fluorescence. Cells were cultured at a density of 0.10/0.20 × 10^5^ per mL in 96 multi-well plates and incubated (24–48 h) in a medium containing the specified treatments. The cell viability assay was performed as described [46]. Briefly, the CyQuant mixture, containing the nuclear dye (CyQuant nuclear stain) and the suppressor of basal fluorescence (background suppressor), was added to the culture medium and incubated for 1 h at 37 °C. Fluorescence was measured at the excitation wavelength of 485 and 530 nm emission. Moreover, the results were expressed as the percentage of fluorescence of the untreated control using a microplate reader (Synergy HT BioTek, Milan, Italy). The experiments were performed in quadruplicate and repeated 3 times. Cells were photographed using the FITC filter (magnification 400×) by an inverted microscope (Axiovert 200 Zeiss, Jena, Germany).

The combination index (CI) values were calculated according to the Chou and Talalay mathematical model for drug interactions, as previously reported [4]. Dose-response curves, dose-effect analysis, and CI for the combination treatment groups were generated with the equations reported by Chou and Talalay using the CompuSyn software (freely available at: www.combosyn.com Accessed on 27 September 2021).

### 4.4. Apoptosis Assays

To measure the caspase-3 enzymatic activity, cells (1.0 × 10^6^/mL) were treated with the indicated molecules for 6 h and then lysed in a lysis buffer (10 mM Hepes, pH 7.4; 2 mM ethylenediaminetetraacetic acid; 0.1% [3-(3-cholamidopropyl) dimethylammonio]-1-propanesulfonate, 5 mM dithiothreitol, 1 mM phenylmethylsulfonylfluoride, 10 μg/mL pepstatin-A, 10 μg/mL apronitin, 20 μg/mL leupeptin). Cell extracts (10 μg) were mixed with the caspase-3 reaction buffer and the conjugated amino-4-trifluoromethyl coumarin (AFC) substrates: Benzyloxycarbonyl-Asp (OMe)-Glu (OMe)-Val-Asp (OMe)-AFC (Z-DEVD-AFC). The samples were incubated at 37 °C for 30 min. Upon proteolytic cleavage of the substrates by caspase-3, the free fluorochrome AFC was detected by a spectrofluorometer multiplate reader (Bio-Tek Instruments) with excitation at 400 ± 20 nm and emission at 530 ± 20 nm. To quantify the enzymatic activities, the AFC standard curve was determined. The caspase-3 specific activity was calculated as nanomoles of AFC produced per min per μg proteins at 37 °C in the presence of saturating substrate concentrations (50 μM) [4].

For apoptotic nuclei staining, after incubation with quercetin, STL1, and their combination for 48 h, cells were collected and centrifuged at 400 g for 5 min, washed in PBS, and stained at 37 °C with the Hoechst solution dissolved in PBS (5 μg/mL) for 30 min. Cells were washed in PBS and apoptotic nuclei were photographed and counted using a fluorescence microscopy (>100 cells/field, 400× magnification) [47].

### 4.5. Autophagy Assays

Autophagy was monitored using the Cyto-ID Autophagy Detection Kit (Enzo Life Science, Euroclone), as described [46]. After incubation for 48 h with quercetin, STL1 or their combination, HG-3 cells (0.10–0.20 × 10^5^/mL) were washed and incubated with the autophagy detection marker (Cyto-ID) and the nuclear dye (Hoechst 33342). Subsequently, cells were rinsed with the assay buffer and photographed using a fluorescence microscope (Zeiss Axiovert 200). Finally, autophagosomes were quantified by normalizing green (Cyto-ID) and blue (Hoechst) fluorescence using a microplate fluorescence reader (Synergy HT BioTek).

To assess the type of autophagy (protective, not protective form) [32] activated by quercetin and STL1 in single treatments or combined at the concentrations indicated in the figures for 48 h, HG-3 cells were pre-incubated for 1 h with two autophagy inhibitors: 25 nM Bafilomycin A1 (BAF) (Abcam, distributed by Prodotti Gianni, Milan, Italy), a (V)-ATPase inhibitor [36] or 3-Methiladenine (3-MA), that can block autophagosome formation via the inhibition of class III PI_3_K [35]. For BAF experiments, Hoechst staining of nuclei was fluorimetrically quantified (340–460 nm excitation-emission wavelength; synergy HT multi-well reader). For the 3-MA experiments, the CyQuant fluorescent dye was used to assess cell viability. The quantity of fluorescence of Hoechst and CyQuant was proportional to the number of cells in each well.

### 4.6. Immunoblotting

After the treatments, cells (usually 1.0–1.5 × 10^6^) were suspended in a lysis buffer consisting of 50 mM Tris-HCl, pH 7.4; 150 mM NaCl; 5 mM ethylenediaminetetraacetic acid; 1% NP-40; 0.5 mM dithiotreitol; 1 mM Na_3_VO_4_; 40 mM NaF; 1 mM Na_4_P_2_O_7_; 34 mM 4-*p*-nitrophenyl phosphate; 10% glycerol; 100 µg/mL phenylmethylsulfonyl fluoride, and a cocktail of inhibitors (Merk Life Science). After measurement of the protein concentration [48], total protein lysates (15–20 μg) were added with a loading buffer (Life-Tecnologies, ThermoFisher Scientific) boiled for 5 min and loaded on a 4–12% pre-cast gel (Novex Bis/Tris Life Technologies) using MOPS (3-(N-morfolin) propanosulfonic) buffer (50 mM MOPS; 50 mM Tris-base; 0.1% SDS; 1 mM EDTA) or MES (2-(N-morpholino) ethanesulfonic acid) buffer (50 mM MES, 50 mM Tris Base, 0.1% SDS, 1 mM EDTA), and a constant voltage (200 V). Blotting of proteins was performed using the hydrophilic polyvinylidene fluoride (PVDF) membrane (Transfer Pack Bio-Rad, Milan, Italy), and TRANS-Blot TURBO System (Bio-Rad), with a constant amperage (2.5 mA) for 7 min at room temperature. After the transfer, membranes were rinsed with T-TBS (0.1% Tween-20; 25 mM Tris; 137 mM NaCl; 2.69 mM KCl, pH 8) and blocked using 5% (*w*/*v*) non-fat dry milk in T-TBS for 1 h at room temperature. The membranes were then incubated for 16 h at 4 °C in the presence of specific antibodies. The primary antibodies used were: Anti-pAKT cod. 4058, anti-LC3 I/II cod. 12741, anti-ATG5 cod. 12994 (Cell Signaling Technologies; Milano, Italy), anti-AKT1 cod. GTX110613 (Genetex distributed by Prodotti Gianni Milano, Italy), and anti-α-tubulin cod. T9026 (Merck). Finally, the PVDF membranes were incubated with the horseradish peroxidase linked secondary antibody against a mouse or rabbit (GE Healthcare, Milan, Italy) and immunoblots were developed using the ECL PRIME detection kit (GE-Healthcare). Band intensities were quantified by measuring the optical density on the Gel Doc 2000 Apparatus (Bio-Rad) and Multi-Analyst Software (Bio-Rad).

## Figures and Tables

**Figure 1 molecules-26-05810-f001:**
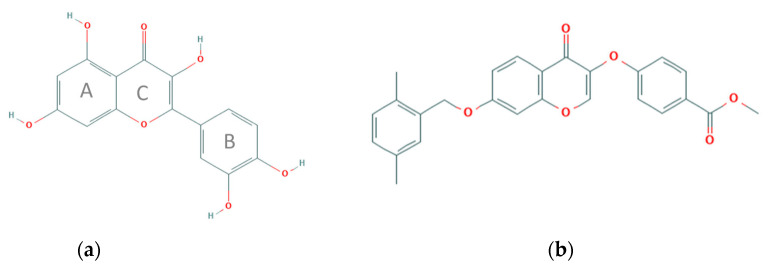
Chemical structures of the compounds investigated in the present study (retrieved and modified from PubChem database, https://pubchem.ncbi.nlm.nih.gov/ Accessed on 27 September 2021). The capacity of quercetin to bind and inhibit kinases involved in the PI3K signaling has been previously reported [4,5,6]. Similarly, a recent study described the structure-activity relationship between the STL1 and AKT1 kinases [7].

**Figure 2 molecules-26-05810-f002:**
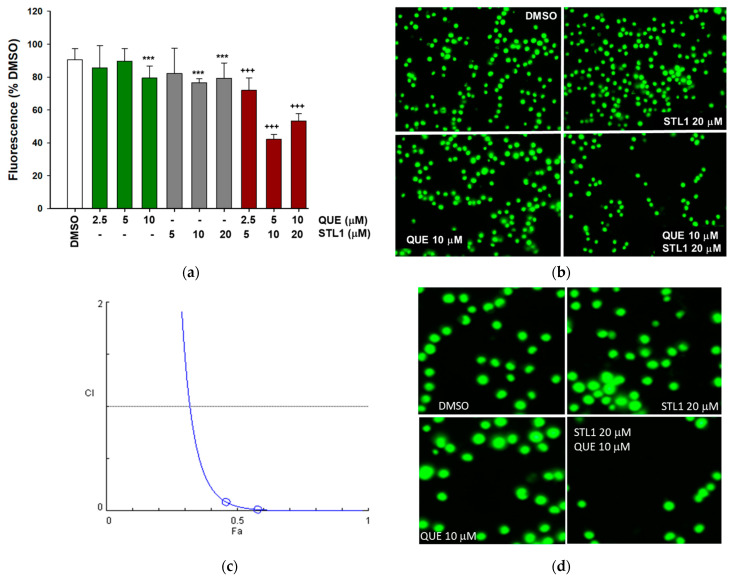
Quercetin and STL1 enhance cell death in HG-3 cells. (**a**) Cells were treated with different doses of STL1, quercetin or their combination, as indicated, for 48 h. Cell viability, measured by the CyQuant assay, is shown as a percentage with respect to the 0.2% DMSO treated control cells, as described in the Materials and Methods section. The white bar (DMSO) represents the cell viability of DMSO-treated vs. untreated control cells to show that DMSO alone does not induce a significant reduction in cell viability. Bar graphs represent the mean of three experiments (±s.e.). Symbols indicate significance: *p* < 0.01 (***) with respect to DMSO; *p* < 0.001 (+++) vs. quercetin and STL1 mono-treatments. (**b**) Microphotographs of HG-3 cells stained with the CyQuant dye after fluorescence reading of the treatments described in (**a**). (**c**) Classic isobologram (for the constant ratio combination design, 2:1 mixture of STL1 and quercetin) obtained a plotting combination index (CI) with the fraction affected (Fa), resulting from CyQuant experiments with HG-3 cells treated for 48 h with three different combinations of STL1 and quercetin. In panel (**d**), we reported a magnification of the four panels shown in panel (**b**) to highlight the morphology of HG-3 cells after the treatment.

**Figure 3 molecules-26-05810-f003:**
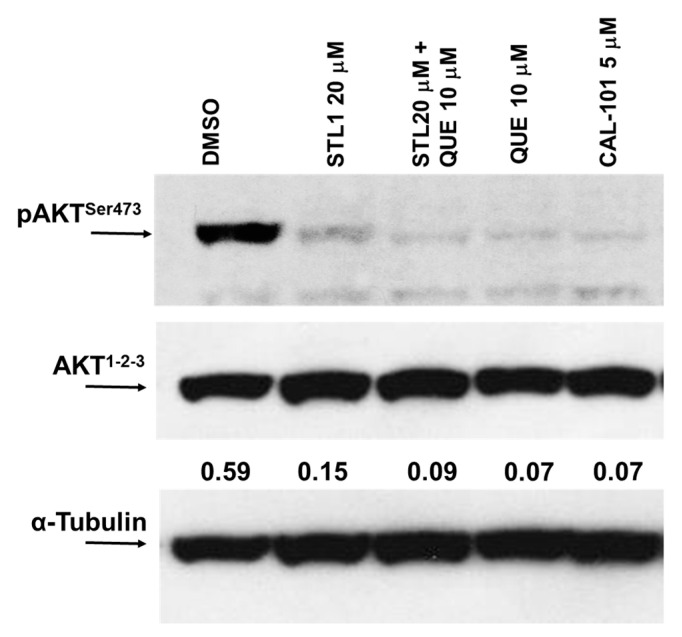
Quercetin enhances STL1 sensitivity in HG-3 through AKT inhibition. Immunoblot of phospho-AKT and AKT protein expressions was performed on HG-3 cells following 60 min of treatment with 10 μM quercetin (QUE), 20 μM STL1, and their association with respect to the DMSO-treated (0.2%) cells. After cell lysis and immunoblotting, membranes were incubated for 16 h at 4 °C with anti-phospho-AKT (pAKT^Ser473^) and re-probed with anti-AKT^1−2−3^ polyclonal antibodies. Finally, the membranes were probed again with an anti-α-tubulin monoclonal antibody. Band intensities were quantified by measuring the optical density and analyzed by the Multi-Analyst Software. The result of the densitometric analysis is reported in the immunoblot and is indicated by the numbers between AKT^1−2−3^ and α-Tubulin panels. Values indicated the AKT^1−2−3^ expression normalized vs. α-tubulin. The image is representative of one out of two independent immunoblots performed.

**Figure 4 molecules-26-05810-f004:**
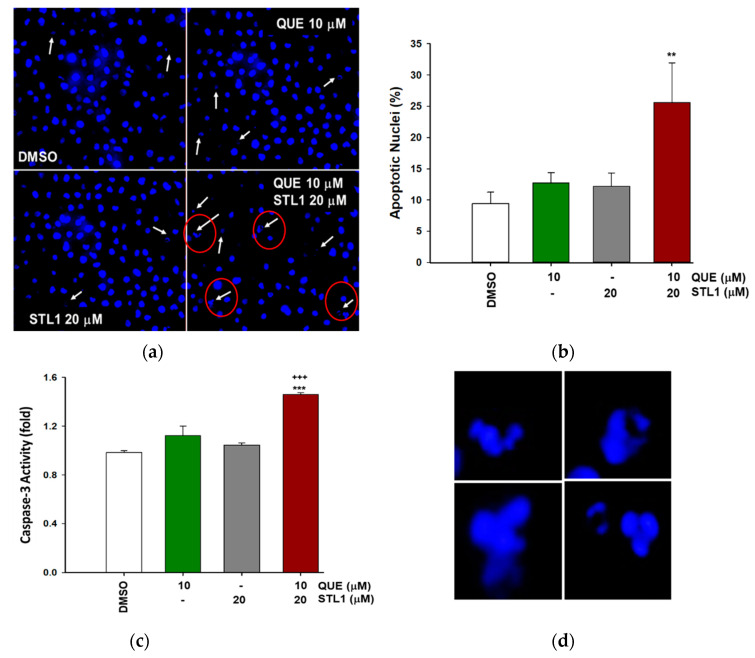
Quercetin and STL1 co-treatment induces apoptosis in HG-3 cells. (**a**,**b**) Hoechst staining of apoptotic nuclei in HG-3 after quercetin (QUE; 10 μM) and STL1 (20 μM) treatments were quantified as described in the Materials and Methods section and photographed using an inverted microscope (Axiovert 200 Zeiss) at 400× magnification. Bar graphs (**b**) represent the mean of three experiments (±s.e.). Symbols indicate significance: *p* < 0.01 (**) with respect to the quercetin and STL1 mono-treatment. (**c**) Proteolytic activation of caspase-3 was measured after 6 h of incubation following the indicated treatments by an enzymatic assay and reported as fold of specific activity with respect to the DMSO treated cells. Bar graphs represent the mean of three experiments (±s.e.). Symbols indicate significance: *p* < 0.001 (***) with respect to quercetin and STL1 mono-treatment or *p* < 0.001 (+++) vs. DMSO-treated cells. In panel (**d**), we reported a magnification of representative examples of apoptotic nuclei indicated by red circles in the lower-right image of panel (**a**) to better highlight their morphology.

**Figure 5 molecules-26-05810-f005:**
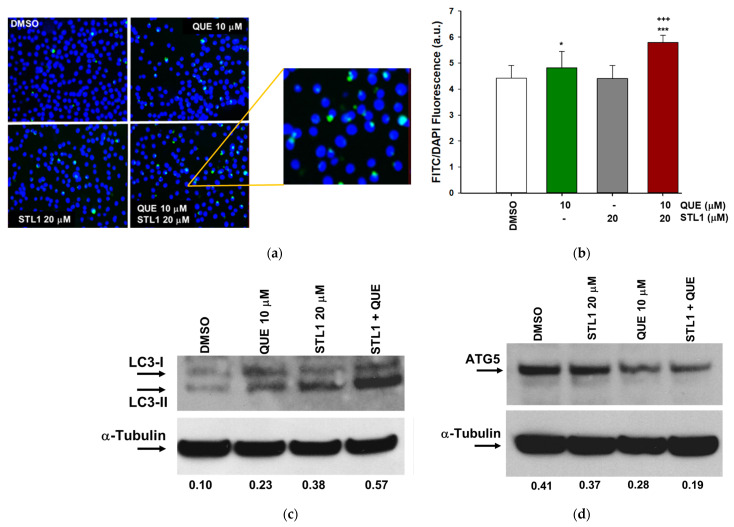
Quercetin and STL1 co-treatment induces autophagy in HG-3 cells. (**a**) Representative images of autophagic vacuoles taken at 400× magnification. In the insert, a magnification of the representative vacuoles is shown. (**b**) Autophagosome quantification expressed as the FITC/DAPI fluorescence ratio (Cyto-ID kit) in HG-3 cells treated for 48 h, as indicated. Bar graphs represent the mean of three experiments (±s.e.). Symbols indicate significance: *p* < 0.001 (***) with respect to the QUE and STL1 mono-treatments; *p* < 0.001 (+++) with respect to DMSO; *p* < 0.05 (*) with respect to DMSO. (**c**,**d**) Immunoblottings of autophagy markers in HG-3 cells treated as indicated for 24 h. Membranes were incubated with anti-LC3 and ATG5 primary antibodies and re-probed with the α-tubulin antibody. Values between panels indicated band intensities normalized with respect to the expression of α-tubulin.

**Figure 6 molecules-26-05810-f006:**
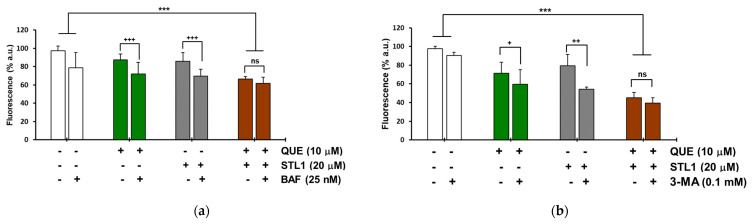
Co-incubation of HG-3 cells with quercetin and STL1 induces non-protective autophagy. Cells were treated with 0.2% DMSO, 10 μM quercetin, 20 μM STL1 or both molecules or pre-treated for 1 h with the autophagy inhibitor. (**a**) BAF (25 nM) or (**b**) 3-MA (0.1 mM) before incubation with quercetin and STL1 at the indicated concentrations. Cell viability was measured after 48 h with (**a**) Hoechst or (**b**) CyQuant staining, as described in the Materials and Methods section. Bar graphs represent the mean ± standard deviation (SD). Symbols refer to significance with respect to the data points indicated by the horizontal square brackets with *p* < 0.001 for (+++) and (***); *p* < 0.01 for (++) and *p* < 0.05 for (+). No significant differences are indicated by (ns).

## Data Availability

Data available on request to the authors.

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
