# Peer review of "STL1, a New AKT Inhibitor, Synergizes with Flavonoid Quercetin in Enhancing Cell Death in A Chronic Lymphocytic Leukemia Cell Line"

_molecules, 2021, doi:10.3390/molecules26195810_

Round 1
Reviewer 1 Report
The manuscript by Russo and coworkers is an interesting piece of scientific evidence in support of the importance of dietary factors or nutraceuticals in disease management. In cases of anti-cancer drugs or antibiotics, the emerging resistance has been increasingly a matter of great concern. Strategies to overcome these resistance for better clinical outcome are essentially required. This group has been actively publishing on anti-cancer agents previously as well and that also includes work on quercetin, a flavonoid. In the current manuscript, the authors have used sound experimental protocols to show a synergistic effect in a combination of quercetin and a novel AKT inhibitor, to induce autophagy in CLL cells. The results are clear and support the conclusion. I found it interesting that the observed cell death were reported at relatively lower concentrations of the agents (10-20 uM), which might have some advantage over the bioavailability issues of flavonoids including quercetin.
A minor query:
As shown in figure 2 (Bar graph-fluorescence representing cell viability), the most effective combination appeared to be 5uM+10 uM (Quercetin + STL1). Why the authors opted for 10uM+20 uM (Quercetin+ STL1) combination for subsequent experiments?
Author Response
Answers-to-Reviewers
Reviewer 1
-The manuscript by Russo and coworkers is an interesting piece of scientific evidence in support of the importance of dietary factors or nutraceuticals in disease management. In cases of anti-cancer drugs or antibiotics, the emerging resistance has been increasingly a matter of great concern. Strategies to overcome these resistance for better clinical outcome are essentially required. This group has been actively publishing on anti-cancer agents previously as well and that also includes work on quercetin, a flavonoid. In the current manuscript, the authors have used sound experimental protocols to show a synergistic effect in a combination of quercetin and a novel AKT inhibitor, to induce autophagy in CLL cells. The results are clear and support the conclusion. I found it interesting that the observed cell death were reported at relatively lower concentrations of the agents (10-20 uM), which might have some advantage over the bioavailability issues of flavonoids including quercetin.
We thank Reviewer-1 for the very positive opinion expressed on our manuscript and for considering it highly significant.
-As shown in figure 2 (Bar graph-fluorescence representing cell viability), the most effective combination appeared to be 5uM+10 uM (Quercetin + STL1). Why the authors opted for 10uM+20 uM (Quercetin+ STL1) combination for subsequent experiments?
Reply-1. Thank you for this keen observation. The main reason for the decision to opt for the combination 10 uM+20 uM quercetin+ STL1 with respect to the alternative lower concentrations (5 uM quercetin +10 uM STL1) resides in the observation that, when we applied the lower combination of concentrations to the assays aimed to assess apoptosis and autophagy (Figures from 3 to 6), it as been hard to obtain consistent and reproducible data. We don’t have a final explanation for this phenomenon that we can bona fide attribute to limited stability of quercetin, STL1, or both in terms of adsorption and/or biotransformation when applied at concentrations too low. We are trying to clarify this aspect. In any case, the determination of the combination index can remove any potential doubt on the dose-dependent effect of the combined treatment as shown in the isobologram of Fig. 2C. For the sake of clarity, we added a new sentence on this issue on the page. 4, lines 140-141 of the revised manuscript.
Reviewer 2 Report
fig2-if data is % increase in DMSO. explain how 1 bar is less than DMSO
photos-it is not possible to see any difference between each photo
fig3: 0.59/0.15/0.09..what do these means? uM?
fig4: photo: it is not possible to identify the apoptotic cells
the discussion could be reduced as in some parts it seems a revision discussion several parts of apoptosis/authpphagy
it is not clear if HG3 cells were obtained from a cell bank or it was "produced" in the lab. please better explain
authors uses caspase 3 as the indicator of apoptosis. and about other caspases?
Author Response
Answers-to-Reviewers
Reviewer-2
-fig2-if data is % increase in DMSO. explain how 1 bar is less than DMSO
Reply-1. Thank you for this observation. Cell viability in Fig. 2 was calculated as a percentage with respect to DMSO-treated cells. The first bar from the left (DMSO) represents cell viability of DMSO-treated vs untreated, control cells to show that DMSO treatment (0.2% v/v) did not induce a significant reduction in cell viability. We clarified this point in the legend of Fig. 2 of the revised manuscript.
-photos-it is not possible to see any difference between each photo
Reply-2. Thank you for this comment. Unfortunately, due to limits in the manuscript format required by the journal, we don’t know how far we can go in enlarging the images. However, to satisfy the reviewer’s request, we prepared inserts for figures 2, 4, and 5 to highlight the most significant details (see changes in the figure legends of the revised manuscript).
-fig3: 0.59/0.15/0.09..what do these means? uM?
Reply-3. As reported in the legend of Fig. 3, “Values between bands indicated protein expression normalized vs a-tubulin”. In other words, those numbers represent the result of the densitometric analysis. We slightly changed the text in the revised manuscript to make this detail more clear (see pg. 5, lines ).
-fig4: photo: it is not possible to identify the apoptotic cells.
Reply-4. See the reply above for Fig. 2
-the discussion could be reduced as in some parts it seems a revision discussion several parts of apoptosis/authpphagy
Reply-5. We agree with this observation and we would like to underline that we did that on purpose for sake of clarity. In fact, some concepts expressed in the Discussion linking functionally apoptosis and autophagy may have appeared too complex for possible generalist readers of Molecules journal and our intent was to help their comprehension by adding more details on the scientific background. Therefore, we kindly request the reviewer to accept the original version as it is without reductions for the reasons expressed above.
-it is not clear if HG3 cells were obtained from a cell bank or it was "produced" in the lab. please better explain
Reply-6. As reported in Materials and Methods section, HG-3 cells were established in the laboratory of Dr. Anders Rosén from a patient affected by CLL and immortalized using the Epstein-Barr virus. The original paper has been published back in 2012 (see the cited reference n. 24). We obtained the cells directly from dr. Rosén although now the cell line is deposited in a cell bank and is commercially available (DSMZ – German Collection of Microorganisms and Cell Cultures). We slightly modified the text of the revised manuscript as reported on pg. 10, lines 364-367.
-authors uses caspase 3 as the indicator of apoptosis. and about other caspases?
Reply-7. We agree with the reviewer that activities of some caspases, e.g. caspase-9 and potentially others are missing. However, a wide range of apoptotic markers are available and cannot be all assayed in the same paper. Here, to explain the role of apoptosis in our experimental model, we measured three markers: caspase-3, apoptotic nuclei, and expression of ATG5 to rationalize the link with autophagy. We think that the use of these markers clearly helped the understanding of this part of the manuscript since we did not receive any negative comments from the three reviewers. As reported in the Discussion, part of the future work will be devoted to measuring the level of Bcl-2 family members in order to assess the capacity of STL-1 to enhance the activity of BH3 mimetics.
Reviewer 3 Report
Revision of the manuscript should:
- provide much more information about STL1 and justify in details its combined treatment with quercetin
- complement the formula of quercetin in figure 1, with formula of STL1, explained in details in the figure legend and explain the respective structure-activity relationships
- not use the term synergy for the combined effects of the substances used and replace it with additive as it is obviously
- be extensively revised considering the above mentioned additive, but not synergistic effects in respect to the activity principles of the substances used
- justify the use of the chosen doses instead of the lower doses that were at least as effective, if not better, as indicated
- explain the above mentioned lack of dose-effect ratio
Author Response
Answers-to-Reviewers
Reviewer-3
-provide much more information about STL1 and justify in details its combined treatment with quercetin
Reply-1. We respectfully think that all the information required is already reported in the original version of the manuscript. See Introduction on page 3 from line 106 about STL1 and page 1 and 2 for the rationale on the use of quercetin as a co-adjuvant agent in CLL supported by coherent literature (see for example refs. 8, 9, 10). In addition, early this year we published on the pages of Molecules a specific paper on STL1 (ref. 23) and, for this reason, we did not think was necessary to repeat in the present manuscript information already published.
-complement the formula of quercetin in figure 1, with formula of STL1, explained in details in the figure legend and explain the respective structure-activity relationships
Reply-2. We respectfully did not understand the present reviewer’s request. We do not see how the two formulas can be complemented. Quercetin and STL1 are two molecules that do not necessarily interact with the same substrates and, in our CLL model, they bind and inhibit different and independent kinases: PI3K and CK2 for quercetin and AKT for STL1. These concepts have been largely described in the Introduction. In addition, as underlined above, the structure-activity relationship of STL1 vs AKT has been recently published by our group (ref. 23). We slightly modified the legend of Figure 1 hoping to partially meet the reviewer’s request.
-not use the term synergy for the combined effects of the substances used and replace it with additive as it is obviously
-be extensively revised considering the above mentioned additive, but not synergistic effects in respect to the activity principles of the substances used
Reply-3. We respectfully disagree with these criticisms and we think that there is not a scientific rationale to substitute the term “synergy” with “additive”. One of the strengths of the present manuscript is the rigorous demonstration of a synergistic effect in the combined treatment quercetin plus STL1. The synergistic, additive, or antagonist effects of drug combinations cannot be determined “by eye” but require the application of methods and protocols largely validated in the pharmacological field. One of these methods, probably one of the most popular, is the Chou and Talalay mathematical model for drug interactions (doi: 10.1016/0065-2571(84)90007-4 and many others) that we rigorously applied here and in other previous papers where we determined the synergistic effects of quercetin in combination with pro-apoptotic drugs. On the opposite, we strongly doubt when we read papers (and unfortunately there are many) that use the terms: “synergistic” or “additive” effects only measuring the "size" of the bars in the graphs in the absence of a pharmacological measurement of these effects. In conclusion, we kindly request the reviewer to reconsider his/her comments in light of our explanation.
-justify the use of the chosen doses instead of the lower doses that were at least as effective, if not better, as indicated
-explain the above mentioned lack of dose-effect ratio
Reply-4. Thank you for this observation. About the lack of dose-effect, we clearly demonstrated that no dose-effect relationship was present in the single treatments (quercetin alone and STL1 alone), while it was the combined treatment that increased the cytotoxic effect of the two compounds as clearly shown by the dose-dependent trend evidenced in Fig. 1. As we already explained in the reply to reviewer-1, the decision to opt for the combination 10 uM+20 uM quercetin+ STL1 with respect to the alternative lower concentrations (5 uM quercetin +10 uM STL1) resides in the observation that, when we applied the lower combination of concentrations to the assays aimed to assess apoptosis and autophagy (Figures from 3 to 6), it as been hard to obtain consistent and reproducible data. We don’t have a final explanation for this phenomenon that we can bona fide attribute to limited stability of quercetin, STL1, or both in terms of adsorption and/or biotransformation when applied at concentrations too low. We are trying to clarify this aspect. In any case, the determination of the combination index can remove any potential doubt on the dose-dependent effect of the combined treatment as shown in the isobologram of Fig. 2C. For the sake of clarity, we added a new sentence on this issue on the page. 4, lines 140-141 of the revised manuscript.